# Development and Application of an Educational-Training Programme for Infection Control Practitioners in Long-Term Care Hospitals

**DOI:** 10.3390/healthcare11040542

**Published:** 2023-02-12

**Authors:** Sun Young Jeong, Og Son Kim

**Affiliations:** 1College of Nursing, Konyang University, Daejeon 35365, Republic of Korea; 2Department of Nursing, Gangseo University, Seoul 07630, Republic of Korea

**Keywords:** infection control, education, long-term care, curriculum

## Abstract

Elderly persons are at risk of infection due to underlying diseases and weak immune systems. All elderly persons do not require hospitalization in LTCHs, even if have chronic illness or weakened immune systems, but they require care in long-term care hospitals (LTCHs) that have well-trained infection control practitioners (ICPs). This study aimed to develop an educational-training programme for ICPs working in LTCHs using the Developing A Curriculum (DACUM) method. Based on the results of the literature review and the DACUM committee workshop, 12 duties and 51 tasks of ICPs were identified. A total of 209 ICPs participated in the survey, rating 12 duties and 51 tasks on a 5-point scale in terms of frequency, importance, and difficulty. An educational-training programme consisting of five modules was developed, focusing on tasks higher than the mean for each of frequency (2.71 ± 0.64), importance (3.90 ± 0.05), and difficulty (3.67 ± 0.44). Twenty nine ICPs participated in a pilot educational-training programme. The mean programme satisfaction level was 93.23 (standard deviation: ±3.79 points) out of 100 points. The average total knowledge and skill scores were significantly higher after the programme (26.13 ± 1.09, 24.91 ± 2.46, respectively) than before the programme (18.89 ± 2.39, 13.98 ± 3.56, respectively) (*p* < 0.001, *p* < 0.001, respectively). This programme will improve the knowledge and skills of ICPs, and thereby contribute to the reduction in healthcare-associated infections in LTCHs.

## 1. Introduction

Korea is an aged society where elderly people aged 65 years or older accounted for 17.6% of the entire population as of 2022 [1]. Among the elderly aged 65 years or older, the prevalence of chronic illnesses diagnosed by a doctor was 89.2%. As the elderly population increased the demands for long-term care service increased accordingly [2].

In Korea, long-term care hospitals (LTCHs) are equipped with at least 30 beds to provide care and healthcare services for patients who require long-term hospitalization for senility or chronic illness, surgical operations, and injuries [3]. The number of such LTCHs in Korea increased drastically from 19 in 2000 to 1467 in 2022 [4].

Elderly persons hospitalized in LTCHs suffer from various underlying problems and are vulnerable to infection due to their relatively weak immune functions [5]. LTCHs are exposed to high risks of healthcare-associated infection (HAI) due to an insufficiency of infection control workforce and infection supervisory systems [6]. In order to address such problems, as of January 2013 the Korean government has obligated all LTCHs nationwide to obtain a medical center certificate including a strengthening criterion regarding infection control among the certification items.

To strengthen infection control, LTCHs need to establish appropriate infection control policies and conduct educational-training programmes for their healthcare workers to implement these policies. There is also a need to monitor whether employees are properly implementing these policies and to provide timely feedback [7]. For these purposes, LTCHs require well-trained infection control practitioners (ICPs) [8].

DACUM stands for “Developing A Curriculum.” The term itself represents the function of an educational development process, but it has been widely used as a job analysis method [9]. This method is systematic in that it effectively helps to identify tasks to be implemented based on a job analysis and the development of an educational process through which trainees can acquire knowledge, attitudes, and skills necessary for their tasks [10]. The DACUM job analysis is effective in developing an educational process for specific tasks because it is based on workshops with the DACUM committee, an expert panel that consists of one trained facilitator and experienced practitioners who have carried out the tasks in the field.

With regard to previous studies on job analysis and education process development by means of DACUM in the area of infection control, Hobbs analyzed Australian infection control professionals’ practice through a 2-day workshop and derived 10 duties [11]. However, Hobbs was limited to job analysis only and presented no further elements of an education process for infection control practitioners based on job analysis results.

As to previous studies on education-training programmes for infection control in long-term care facilities, Daly et al. reported that after the 2-day basic training programme developed by the Nebraska infection control network, knowledge and performance were improved among 266 ICPs in long-term care facilities [12]. In a survey conducted among 142 individuals who completed infection control training programs in long-term care facilities or medium and small sized hospitals, it turned out that most of them handled multiple tasks such as quality management, employee health management, general nursing, administration, and safety management [13]. A total of 96% of the trainees continued to carry out infection control tasks, and they reported that the education programme was useful for their practical tasks regarding surveillance/data analysis, resistant organism control, and isolation/standard precautions [13]. However, although the contents of the infection control education programme were generally clarified through these two studies [12,13], specific procedures for developing the curriculum were not drawn.

Therefore, the purpose of the study was to develop an infection control educational-training programme for ICPs working in LTCHs based on a job analysis that utilizes the DACUM method, with the aim of improving the work performance of ICPs in LTCHs.

## 2. Materials and Methods

### 2.1. Research Design

This is a descriptive study that analyzed the duties of ICPs in LTCHs and developed an educational-training programme by means of the DACUM method.

### 2.2. Study Subject and Data Collection

In order to analyze the duties of ICPs in LTCHs, a six-member DACUM committee was organized with experts including two infection control nurses with at least 10 years of professional experience, two ICPs working in LTCHs, and two nursing professors who conducted a Master’s degree course on infection control. By means of the simplified DACUM method [10], the researcher examined, in a preliminary step, the duties of ICPs in long-term care facilities based on studies on the duties of ICPs in healthcare facilities [7,14]. A one-day workshop was then conducted on 2 September 2016 to identify the duties and tasks of ICPs in LTCHs with the six-member DACUM Committee.

One DACUM facilitator and two DACUM coordinators brought in to assist with the analysis conducted the DACUM job analysis 6-h workshop. The total workshop time was 8 h. The facilitator had experience in defining the task of swallowing rehabilitation using DACUM, developing a ‘swallowing rehabilitation’ course for occupational therapists related to swallowing rehabilitation, and publishing the article. In accordance with DACUM’s procedures, the DACUM facilitator provided orientation on the DACUM job analysis to ICPs in LTCHs and obtained written informed consent from six participants. The participants clarified the job definition of the ICP in LTCHs, and analyzed and named their duties and tasks. The coordinators’ role was to assist the facilitator by recording duties and tasks or creating DACUM charts. The workshop was conducted in a comfortable atmosphere with equal treatment regardless of status and age so that participants could participate freely without using references. Duties and tasks are recorded and organized after participants agree through discussion.

In order to verify the identified duties and tasks, a list of 1408 LTCHs was received from the Korean Association of Geriatric Hospitals. From this list, through convenience sampling, 10 to 20 LTCHs were selected for each of 17 metropolitan local governments in Korea, and less than 100 beds, 100 to less than 200 beds, and more than 200 beds were evenly distributed by bed size, resulting in a total of 250 LTCHs being selected.

A survey of 250 ICPs was conducted via e-mail on the frequency, importance, and difficulty of each task from 30 September to 7 November 2016, and 209 responses were received. Based on the survey results, a 4-day educational-training programme was developed that included five modules, focusing on tasks with above-average frequency, importance, and difficulty scores. To verify the effectiveness of the educational-training programme, a pilot study was conducted. The sample size was estimated using the G*power 3.1.1 software program. Sample size estimation was based on the following Wilcoxon signed-rank test criteria: statistical power of 0.80, significance level of 0.05, effect size of 0.50, and a one-sided test. Therefore, we needed a sample size of 28 participants. Allowing for attrition, we recruited a total of 30 participants among the ICPs who responded to the survey sent by e-mail and who wished to attend the programme.

ICPs who completed the 4-day programme from 8 to 11 February 2017 had their post-program satisfaction level measured as the process evaluation immediately after completing the programme. To evaluate the effectiveness before and after the programme, the level of knowledge, skills, and awareness of infection control were measured immediately after completing the programme.

### 2.3. Research Tool

#### 2.3.1. The Frequency, Importance, and Difficulty of the Tasks of ICPs in LTCHs

In this study, the task frequency, importance, and difficulty of ICPs in LTCHs were investigated in order to derive tasks requiring the development of educational-training programmes. To investigate the frequency, importance, and difficulty of the tasks of ICPs in LTCHs, this study utilized a task verification questionnaire that consisted of questions regarding tasks of ICPs that were identified through a workshop of DACUM committee members. The degree of frequency, importance, and difficulty was measured for each question on a 5-point scale ranging from 1 to 5 (with higher points indicating greater frequency, importance, and difficulty). In this study, the Cronbach’s alpha indicating the reliability of each question’s degree of frequency, importance, and difficulty was 0.963, 0.971, and 0.956, respectively.

#### 2.3.2. Programme Satisfaction

To evaluate the process of the educational-training programme, participants’ satisfaction with education was investigated. A tool developed by Jung and Lee [15] to measure the college educational level of satisfaction was utilized in this study. This tool consisted of 15 questions including educational themes, contents, instructors, methods, and materials. Each question was given points on a 5-point scale range from 1 to 5 (higher points indicates more satisfaction). The tool’s reliability had a Cronbach’s alpha = 0.899.

#### 2.3.3. Knowledge and Skills

To evaluate the effectiveness of the educational-training programme on learners, knowledge and skills were investigated. In order to determine whether the learning goals of each module were achieved, 30 questions each about knowledge and skills were developed. Each correct answer was given 1 point, and each incorrect or no answer was given 0 points. The perfect score of the knowledge questions was 30 points, and that of skill questions was 30 points.

#### 2.3.4. Awareness of Infection Control

In order to evaluate the effectiveness of the educational-training programme, the infection control awareness tool developed by Hong and Park [16] was used. The two questions regarding a catheter that were not applicable to ICPs in LTCHs were excluded, while 35 questions in total including hand hygiene, employee safety, intravascular catheter infection control, urinary tract infection control, pneumonia control, isolation, as well as disinfection and sterilization management were included. Each answer was measured on a 5-point scale. Cronbach’s alpha indicating the reliability of the tool before and after the education programme was 0.962 and 0.965, respectively.

### 2.4. Data Analysis

Each task’s degree of frequency, importance, difficulty, and educational level of satisfaction were analyzed using descriptive statistics and were reported as frequency, percentage, average, and standard deviation. Differences before and after the education program in scores of knowledge, skill, and awareness of the importance of infection control, were analyzed by means of a Wilcoxon signed-rank test. A data analysis was conducted using SPSS (IBM SPSS Statistics for Windows, Version 18.0., IBM Corp.: Armonk, NY, USA)

## 3. Results

### 3.1. Job Analysis: Duties and Tasks of ICPs in LTCHs

Based on the results of the DACUM committee workshop, 12 duties and 51 tasks of ICPs were identified (Table 1). Specifically, the following 12 duties were derived: preparing policies and guidelines; healthcare associated infection (HAI) surveillance; planning and evaluating the infection control programme; hand hygiene promotion; medical instrument disinfection and sterilization management; isolation precaution to patients with infectious disease including reportable diseases; infection control of each strain; infection control for each site; infection control for healthcare workers (HCWs); environmental infection control; consultation and communication with internal and external departments; and self-development. One to seven tasks for each duty were included.

### 3.2. Task Verification

The degrees of frequency, importance, and difficulty of each task of infection control were analyzed among 209 practitioners in long-term care facilities. The analysis results are presented in Table 1. The degree of frequency was 2.71 ± 0.64 on average in the 5-point scale. The following were tasks of a high degree of frequency, in order: laundry management (3.92 ± 0.82), medical waste management (3.91 ± 1.01), hand hygiene practice investigation (3.63 ± 0.92), environmental cleaning (3.61 ± 0.87), hand hygiene promotion activity (3.57 ± 0.95), urinary tract infection control (3.54 ± 1.06), disposable goods recycling management (3.44 ± 1.22), clean/contaminated zone classification (3.43 ± 1.00), preparation for certification evaluation (3.32 ± 1.03), investigation of sterilization methods, and suggestions (3.26 ± 1.00).

The score of importance was 3.90 ± 0.05 on average in the 5-point scale. The following were tasks of a high level of importance, in order: scabies control (4.32 ± 0.72), hand hygiene practice investigation (4.27 ± 0.68), hand hygiene promotion activity (4.27 ± 0.68), disposable goods recycling management (4.25 ± 0.68), tuberculosis control (4.25 ± 0.71), medical waste management (4.25 ± 0.70), urinary tract infection control (4.24 ± 0.70), pneumonia control (4.22 ± 0.70), legal epidemic patient care (4.18 ± 0.74), and VRE infection control (4.17 ± 0.76).

The score of difficulty was 3.67 ± 0.44 on average in the 5-point scale. The following were tasks of a high level of difficulty, in order: preparation for certification evaluation (4.07 ± 0.75), health-care associated infection (HAI) result analysis (3.89 ± 0.83), scabies control (3.89 ± 0.74), VRE infection control (3.89 ± 0.71), infection outbreak investigation and reporting (3.88 ± 0.87), health-care associated infection (HAI) result evaluation and reporting (3.86 ± 0.83), infection control planning (3.84 ± 0.77), index management (3.83 ± 0.86), health-care associated infection (HAI) investigation (3.83 ± 0.82), and tuberculosis control (3.83 ± 0.73).

In summary, although laundry management, advice on medical waste and hand hygiene were performed frequently, the scores indicating their levels of difficulty were lower. However, despite the low frequency of preparation for certification evaluation, the degree of difficulty was very high.

### 3.3. Key Task Selection for Development of Educational-Training Programme

For educational-training programme development, among the tasks of the ICP in LTCHs, tasks with frequency (average 2.71), importance (average 3.90), and difficulty (average 3.67) scores above average were selected. These tasks included the following eight tasks: selecting disinfectant, identifying and advising how to disinfect medical equipment, influenza control, scabies control, bacteremia control, pneumonia control, urinary tract infection control, and preparing for the Korean accreditation programme for healthcare organizations (Table 1). Although the frequency of the methicillin-resistant *Staphylococcus aureus* (MRSA) infection control task was lower than the average, researchers selected it as a key task for the development of an educational-training programme, considering the increased isolation of multidrug-resistant organisms (MDROs) in LTCHs in Korea [17].

The elements for each key task were derived from the workshop involving the DACUM committee members. Several elements of the nine key tasks are shown in Table 2.

### 3.4. Educational-Training Programme Development

The knowledge and skills required for each element of each task were analyzed, and the expected subject areas from the nine key tasks were classified into five modules of the educational-training programme (Figure 1; Table 3). For each module, the lecture contents and materials, practice contents and materials were developed. Lecture and practice materials included the policy plans, lecture plans, monitoring feedback forms, and report forms.

### 3.5. Educational-Training Programme Operation

The five educational-training program modules were conducted over a period of 22 h during the 4-day educational process from 8 to 11 February 2017 (Table 3). Each module was conducted through lectures and practical sessions using developed lecture and practice materials. Of the 30 ICPs, 29 completed the 4-day program.

### 3.6. Educational-Training Programme Evaluation

As a result of the process evaluation, the programme satisfaction level was 93.23 ± 3.79 points on average on a 100-point scale (Table 4). All of the modules had a minimum of 90 points on the satisfaction level. The average scores of infection control knowledge before and after the programme were 18.89 ± 2.39 and 26.13 ± 1.09 points, respectively, indicating a significantly higher score after the education (z = –4.70, *p* < 0.001). The average score of infection control skills before and after the programme (13.98 ± 3.56 and 24.91 ± 2.46 points, respectively) was also significantly higher after the education (z = −4.70, *p* < 0.001). However, the average scores of awareness of infection control on a 175-point scale (168.90 ± 8.74 and 171.10 ± 7.36 points, respectively) showed no statistically significant difference (z = –1.77, *p* = 0.077) (Table 4). In summary, programme satisfaction was higher than 90 points, and knowledge and skills improved after the programme.

## 4. Discussion

This present study developed an educational-training programme for ICPs working in LTCHs based on a job analysis that utilized the DACUM method. Following the analysis of the job of ICPs in LTCHs in Korea, 12 duties and 51 tasks were identified. In general, these duties and tasks were similar to those reported in previous studies on the activities of ICPs in acute healthcare facilities in Korea [18], guidelines for infection control in long-term care facilities in Japan [5], and core competencies of infection control nurse specialists in Hong Kong [7], as well as on the roles and competencies of the Association for Professionals in Infection Control and Epidemiology (APIC) infection preventionists [19]. When the findings of this study were compared with those of Hobbs [11], who similarly derived areas of duties of ICPs in Australia by applying the DACUM method, the following additional duties were identified: hand hygiene promotion, medical instruments disinfection and sterilization management, environmental infection control, and preparation for certification evaluation. As the main role of ICPs was changed recently from “control” to “prevention”, it was thought that their duties such as programme planning, budget management, and preparation for certification were emphasized, in addition to main duties such as surveillance, reporting, and education [20].

In this study, key tasks for the development of educational-training programmes were selected, including eight tasks with higher than average frequency, importance, and difficulty, and MRSA infection control, which had a lower than average frequency, but high importance and difficulty. The prevalence of MDROs in long-term care facilities varies by continent, and it is known that the prevalence of MRSA in Asia is higher than in North America [21]. MDROs including MRSA are increasing due to antibiotics used to prevent secondary bacterial infection of Coronavirus disease 2019 (COVID-19) during the COVID-19 epidemic [22]. Accordingly, the importance of MDROs, including MRSA screening tests and proactive isolation is being emphasized in LTCHs [23].

In particular, the fact that >78.5% of patients in long-term care facilities have urinary catheters and 93% receive influenza vaccines, indicates the importance of urinary tract infection and influenza control in long-term care facilities [24]. Therefore, the five modules developed in this study (hand hygiene and safe injection practice, disinfection and environment control, risk assessment and multidrug-resistant strain control, urinary tract infection control, and respiratory infection and visitor control) are effective for competency enhancement of ICPs in LTCHs.

In this study, the educational-training programme consisting of lectures and practice was conducted over 4 days, for a total of 22 h. This programme is of significance in that it included both lectures and practical sessions as a reflection of the needs of ICPs in Korean LTCHs [6]. ICPs should apply a variety of teaching methods, including workshops, team-based learning, problem-based learning, on-site training, and simulations to help healthcare workers put their infection control knowledge into practice [14,25]. This programme will help ICPs apply such various educational methods in their LTCHs.

The effect of the educational-training programme developed in this study was measured through process evaluation and effectiveness evaluation. The level of satisfaction of subjects with the programme measured through the process evaluation was high. The effectiveness evaluation was used to measure the effect of the programme on subjects’ knowledge, infection control awareness, and skills. In this study, the scores of knowledge and skills were higher after the programme.

However, there was no difference in infection control awareness before and after education. This was thought to be because the infection control awareness score of the subjects of this study was relatively high even before the programme, and there was not enough time to bring about the additional awareness changes. Therefore, in future studies, it is necessary to measure the change in awareness after the passage of time after educational-training programmes. This study developed the educational-training programme and tested the programme only among 29 individuals as a pilot project. Thus, a future study needs to include more subjects.

This study is of significance in that it developed a programme systematically based on the duties of ICPs in LTCHs by means of the DACUM method in order to improve infection control in LTCHs where human resources and infrastructure, such as facilities and devices, are inadequate in comparison to that of acute healthcare facilities. Nonetheless, it is necessary to include more ICPs in the DACUM panel to verify their duties more specifically. Since it takes more time to change awareness than merely acquire knowledge, the effectiveness of the programme needs to be re-evaluated over time. In addition, of the 51 tasks, 50 had a difficulty level of between 3 and 4 points, indicating that most of the infection control tasks were recognized as slightly difficult by infection control practitioners. This is consistent with evidence showing that the knowledge evaluation score prior to the operation of the educational-training programme developed based on task analysis was slightly lower, between 2 and 3 points, in most items. Therefore, efforts to increase knowledge and competence through educational-training programmes are necessary. And follow-up studies to verify the infection control performance and the HAI rate as indicators for evaluating the effectiveness of educational-training programmes should be continued.

## 5. Conclusions

An infection control educational-training programme was developed systematically in this study for ICPs working in LTCHs by means of the DACUM job analysis method and a thorough application of the educational process development procedures. The process evaluation and effectiveness evaluation conducted before and after the educational-training programme proved the outstanding training programme performance. Particularly, regarding knowledge and skills, the scores after programme implementation were significantly high for all five modules. The educational satisfaction level was as high as 90 points in every module. Therefore, this programme can be used for ICPs of various LTCHs in Korea, where LTCHs are rapidly increasing. Through this programme, it is expected that the knowledge, awareness, and skills of infection control in ICPs in LTCHs will be improved, thereby improving job competence. Additionally, the policy plans, lecture plans, monitoring feedback forms, and report forms developed for each module in this educational-training programme can be applied directly to infection control work in LTCHs, being expected to contribute to improved infection control practices in such facilities.

This study was unable to evaluate whether an educational-training programme improved infection control activities and reduced healthcare-associated infection rates in LTCHs due to study period constraints. Therefore, we suggest a study on the evaluation of results according to the application of educational-training programmes in the future.

## Figures and Tables

**Figure 1 healthcare-11-00542-f001:**
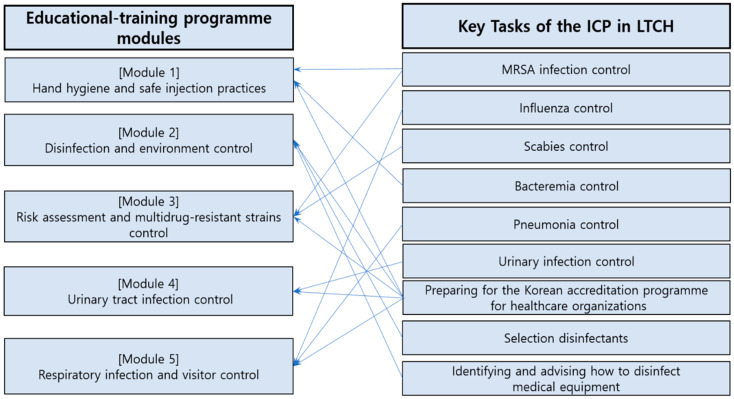
Educational-training programme modules based on the identified key tasks using the DACUM method. This figure shows that five modules were developed to teach the knowledge and skills needed to perform nine key tasks. For example, the “risk assessment and multidrug-resistant strains control” module was developed to teach knowledge and skills required for “MRSA infection control”, “scabies control”, and “preparing for the Korean accreditation program for healthcare organizations.” ICP, Infection Control Practitioner; LTCH, Long-term Care Hospital; MRSA, methicillin-resistant *Staphylococcus aureus*; DACUM, Developing A Curriculum.

**Table 1 healthcare-11-00542-t001:** Degree of Frequency, Importance, and Difficulty of Duties and Tasks (N = 209).

Duties	Tasks	Frequency ^a^		Importance ^b^		Difficulty ^c^	
		M ± SD	Rank	M ± SD	Rank	M ± SD	Rank
1. Preparing policies and guidelines	1-1. Developing infection control guidelines and policies	2.51 ± 0.97	30	3.76 ± 0.82	32	3.82 ± 0.80	11
	1-2. Revising infection control guidelines and policies	2.58 ± 0.96	27	3.78 ± 0.79	31	3.78 ± 0.80	14
2. Healthcare associated infection (HAI) surveillance	2-1. Investigating HAI control practice	2.45 ± 1.05	34	3.68 ± 0.81	36	3.83 ± 0.82	9
	2-2. Analyzing results of HAI control	2.41 ± 1.05	36	3.66 ± 0.81	41	3.89 ± 0.83	2
	2-3. Evaluating and reporting HAI control results	2.46 ± 1.05	32	3.67 ± 0.80	40	3.86 ± 0.83	6
	2-4. Investigating and reporting infection outbreaks	2.34 ± 1.11	37	3.68 ± 0.87	36	3.88 ± 0.87	5
3. Planning and evaluating an infection control programme	3-1. Planning an infection control programme (including budgets)	2.24 ± 0.98	42	3.68 ± 0.81	36	3.84 ± 0.77	7
	3-2. Establishing an infection control report system	2.52 ± 0.97	29	3.68 ± 0.82	36	3.69 ± 0.75	25
	3-3. Index management (investigation, analysis, evaluation)	2.26 ± 0.93	41	3.57 ± 0.90	45	3.83 ± 0.86	8
	3-4. Conducting infection control promotion events (The Day of Infection Control, etc.)	1.94 ± 0.94	49	3.34 ± 0.92	49	3.66 ± 0.90	29
	3-5. Reporting project results and preparing a report	2.09 ± 0.98	48	3.44 ± 0.91	48	3.66 ± 0.85	30
4. Hand hygiene promotion	4-1. Investigating the level of hand hygiene practice	3.63 ± 0.92	3	4.27 ± 0.68	2	3.62 ± 0.77	40
	4-2. Hand hygiene promotion activity	3.57 ± 0.95	5	4.27 ± 0.68	2	3.63 ± 0.78	38
5. Medical instrument disinfection and sterilization management	5-1. Selecting disinfectants ^†^	3.21 ± 0.95	12	4.09 ± 0.67	16	3.68 ± 0.72	28
	5-2. Identifying and advising how to disinfect medical equipment ^†^	3.15 ± 0.96	13	4.03 ± 0.70	22	3.70 ± 0.69	24
	5-3. Disposable goods recycling management	3.44 ± 1.22	7	4.25 ± 0.68	4	3.63 ± 0.79	37
	5-4. Sterilization disinfection method investigation and suggestion	3.26 ± 1.00	10	4.03 ± 0.72	22	3.66 ± 0.70	32
6. Isolation precautions to patients with infectious diseases, including reportable diseases	6-1. Checking isolation precautions depending on the symptoms	2.71 ± 1.21	20	4.03 ± 0.79	22	3.64 ± 0.76	35
	6-2. Isolating patients depending on the symptoms	2.59 ± 1.17	25	4.01 ± 0.78	27	3.62 ± 0.84	39
	6-3. Isolation room management	2.65 ± 1.24	23	4.05 ± 0.80	20	3.68 ± 0.82	27
	6-4. Wearing personal protective equipment	2.79 ± 1.23	19	4.12 ± 0.71	15	3.64 ± 0.80	33
	6-5. Reporting reportable diseases	2.14 ± 1.27	45	4.16 ± 0.75	11	3.66 ± 0.84	31
	6-6. Transferring patients with reportable diseases to another hospital	2.16 ± 1.27	44	4.18 ± 0.74	9	3.76 ± 0.84	16
7. Infection control of each strain	7-1. MRSA infection control ^†^	2.62 ± 1.31	24	4.13 ± 0.76	14	3.81 ± 0.70	13
	7-2. VRE infection control	2.58 ± 1.35	26	4.17 ± 0.76	10	3.89 ± 0.71	4
	7-3. Other antimicrobial resistant bacteria control (CRE, etc.)	2.45 ± 1.27	33	4.08 ± 0.84	17	3.82 ± 0.75	12
	7-4. Tuberculosis control	2.69 ± 1.34	21	4.25 ± 0.71	4	3.83 ± 0.73	10
	7-5. Influenza control ^†^	2.85 ± 1.23	17	4.07 ± 0.80	18	3.72 ± 0.79	20
	7-6. *C. difficile* control	2.46 ± 1.25	31	3.95 ± 0.88	30	3.73 ± 0.80	19
	7-7. Scabies control ^†^	3.02 ± 1.33	16	4.32 ± 0.72	1	3.89 ± 0.74	3
8. Infection control for each site	8-1. Bacteremia control ^†^	3.09 ± 1.17	15	4.15 ± 0.71	13	3.70 ± 0.75	23
	8-2. Pneumonia control ^†^	3.23 ± 1.15	11	4.22 ± 0.70	8	3.78 ± 0.74	15
	8-3. Urinary tract infection control ^†^	3.54 ± 1.06	6	4.24 ± 0.70	7	3.72 ± 0.75	21
9. Infection control for healthcare workers (HCWs)	9-1. Development and operation of an HCWs infection control programme	2.34 ± 1.03	38	3.73 ± 0.76	33	3.75 ± 0.75	17
	9-2. Development and operation of a vaccination programme	2.57 ± 1.01	28	3.72 ± 0.76	34	3.60 ± 0.75	41
	9-3. Management of HCWs exposure to infectious diseases	2.67 ± 1.05	22	3.96 ± 0.72	29	3.69 ± 0.72	26
	9-4. Blood and body fluid exposure management	2.83 ± 1.11	18	4.02 ± 0.72	26	3.64 ± 0.77	34
10. Environmental infection control	10-1. Air and water quality management	2.28 ± 1.01	40	3.70 ± 0.78	35	3.71 ± 0.76	22
	10-2. Infection control in construction	2.11 ± 0.98	47	3.61 ± 0.82	43	3.73 ± 0.81	18
	10-3. Clean/contaminated area distinction	3.43 ± 1.00	8	4.03 ± 0.70	22	3.46 ± 0.82	44
	10-4. Advice on environmental cleaning	3.61 ± 0.87	4	4.04 ± 0.67	21	3.36 ± 0.75	46
	10-5. Environmental disinfection method investigation and suggestion	3.14 ± 1.00	14	3.99 ± 0.71	28	3.64 ± 0.70	36
	10-6. Laundry management	3.92 ± 0.82	1	4.16 ± 0.66	11	3.46 ± 0.75	45
	10-7. Advice on medical waste management	3.91 ± 1.01	2	4.25 ± 0.70	4	3.52 ± 0.80	43
11. Consultation and communication with internal and external departments	11-1. Consultation on infection control	2.23 ± 0.96	43	3.62 ± 0.83	42	3.35 ± 0.73	48
	11-2. Participating in hospital meetings	2.43 ± 1.09	35	3.53 ± 0.86	46	3.28 ± 0.81	49
	11-3. Organizing and participating in the infection control committee	2.34 ± 1.02	39	3.52 ± 0.88	47	3.35 ± 0.78	47
	11-4. Collaboration with public health centers	1.92 ± 0.9	50	3.19 ± 1.00	50	3.23 ± 0.88	51
	11-5. Publishing infection control newsletters	1.57 ± 0.89	51	3.13 ± 1.00	51	3.58 ± 0.95	42
	11-6. Preparing for the Korean accreditation programme for healthcare organizations ^†^	3.32 ± 1.03	9	4.06 ± 0.73	19	4.07 ± 0.75	1
12. Self-development	12-1. Participating in education programme and seminars outside the hospital	2.14 ± 0.99	46	3.61 ± 0.94	43	3.25 ± 0.88	50

^a^ 5-point scale, 1 = rarely, 2 = sometime, 3 = neutral, 4 = often, 5 = very often; ^b^ 5-point scale, 1 = very low, 2 = low, 3 = neutral, 4 = high, 5 = very high; ^c^ 5-point scale, 1 = very low, 2 = low, 3 = neutral, 4 = high, 5 = very high; ^†^ Key tasks = tasks selected for the development of educational-training programmes. M, mean; SD, standard deviation; MRSA, methicillin-resistant *Staphylococcus aureus*; VRE, vancomycin-resistant enterococci; CRE, carbapenem-resistant Enterobacteriaceae; *C. difficile*, *Clostridioides difficile.*

**Table 2 healthcare-11-00542-t002:** Elements of key tasks.

Key Task	Elements of Key Task
Selecting disinfectants	Categorizing the infection risks of the object (goods, environmental surface, etc.) to be disinfected Selecting disinfectants according to disinfection level
	Performing procedures to introduce disinfectants
Identifying and advising how to disinfect medical equipment	Classifying the infection risks of equipment Determining the level of disinfection according to the infection risks of the equipment to be disinfected Identifying disinfectant types and disinfection methods and procedures according to disinfection level Providing feedback to the department or staff about the disinfection method identified
MRSA infection control	Establishing guidelines (policies) for multidrug-resistant strain control Preparing sickrooms for patients with multidrug-resistant strains Selecting resources for multidrug-resistant strain control Education for employees on multidrug-resistant strain control Monitoring employees on the compliance with guidelines and precautions for isolation Collecting feedback on monitoring results Preparing outbreak reports Preparing monitoring reports Activity for multidrug-resistant strain control promotion
Influenza control	Establishing guidelines (policies) for influenza control
	Education for employees on influenza control
	Monitoring influenza outbreaks Managing patient and employee exposure to influenza Monitoring compliance with droplet precautions
	Feedback monitoring results to departments and employees
	Preparing outbreak reports and monitoring reports if needed
Scabies control	Establishing guidelines (policies) for scabies control
	Education for employees on scabies control
	Monitoring the occurrence of scabies Managing patient and employee exposure to scabies
	Monitoring compliance with contact precautions
	Preparing outbreak reports and monitoring reports if needed
Bacteremia control	Establishing guidelines (policies) for bloodstream infection control
	Determining healthcare-related bloodstream infection surveillance standards and methods Training procedures and methods to prevent bacteremia Monitoring compliance with guidelines (policies) Preparing outbreak reports and monitoring reports if needed Activity for infection control promotion (ex: bundle approach)
Pneumonia control	Establishing guidelines (policies) for pneumonia control
	Determining healthcare-related pneumonia surveillance standards and methods Training procedures and methods to prevent pneumonia Monitoring compliance with guidelines (policies) Preparing outbreak reports and monitoring reports if needed Activity for infection control promotion (ex: bundle approach)
Urinary tract infection control	Establishing guidelines (policies) for urinary tract infection (UTI) control
	Determining healthcare-related UTI surveillance standards and methods Training procedures and methods to prevent UTI Monitoring compliance with guidelines (policies) Preparing outbreak reports and monitoring reports if needed Activity for infection control promotion (ex: bundle approach)
Preparing for the Korean accreditation program for healthcare organizations	Identifying the accreditation criteria for infection control Ensuring that infection control policies meet accreditation criteria Identifying the infrastructure required to implement infection control policies Monitoring compliance with infection control policies Preparing reports and evidence data for accreditation

MRSA, methicillin-resistant *Staphylococcus aureus.*

**Table 3 healthcare-11-00542-t003:** Educational-training programme for infection control practitioners in long-term care hospitals based on the DACUM method.

Module Name	Schedule		Theme
Hand hygiene and safe injection practice	1st Day	30 min	Pre-test and introduction to the education process
		100 min	Hand hygiene theories and practice
		90 min	Hand hygiene practice evaluation and promotion activity
		90 min	Safe injection theories and practice
		60 min	Safe injection practice evaluation and promotion activity
Disinfection and environment control	2nd Day	100 min	Understanding of disinfection and sterilization/Standards for disinfection and environment control
		90 min	Principles of disinfection and environment control
		60 min	Practical application of disinfection and environment control
		60 min	Activity for disinfection and environment control enhancement
Risk assessment and multidrug-resistant strains control	3rd Day	100 min	Risk assessment methodology
		90 min	Risk assessment application and reporting
		90 min	Epidemiology and control of multidrug-resistant strain
		60 min	Practical application of multidrug-resistant strain infection, performance evaluation, and promotion activity
Urinary tract infection control Respiratory infection and visitor control	4th Day	100 min	Urinary tract infection epidemiology and infection control
		90 min	Urinary tract infection control indexes and reporting
		90 min	Aspiration pneumonia infection control intervention and index management
		60 min	Influenza infection control intervention and index management/visitor management

DACUM, Developing A Curriculum.

**Table 4 healthcare-11-00542-t004:** Evaluation of educational-training programme for infection control practitioners in long-term care hospitals based on the DACUM method (N = 29).

	Before Programme	After Programme	z(*p*) ^*^
	M ± SD	M ± SD	
Process evaluation			
Programme satisfaction level ^†^		93.23 ± 3.79	
Hand hygiene and Safe injection practices		90.57 ± 10.05	
Disinfection and environmental control		92.55 ± 10.06	
Risk assessment ang multidrug-resistant strains control		95.91 ± 6.89	
Urinary tract infection control		93.38 ± 9.05	
Respiratory infection and visitor control		93.47 ± 10.40	
Effectiveness evaluation			
Knowledge ^‡^	18.89 ± 2.39	26.13 ± 1.09	−4.70 (<0.001)
Hand hygiene	2.50 ± 0.56	2.89 ± 0.40	−2.96 (0.003)
Safe injection practices	4.46 ± 1.02	5.00 ± 0.00	−3.06 (0.002)
Disinfection and environmental control	3.21 ± 1.05	4.79 ± 0.49	−4.34 (<0.001)
Risk assessment ang multidrug-resistant strains control	3.83 ± 1.17	5.00 ± 0.00	−3.89 (<0.001)
Urinary tract infection control	3.43 ± 0.94	4.63 ± 0.47	−4.06 (<0.001)
Respiratory infection and visitor control	1.47 ± 1.02	3.82 ± 1.00	−4.47 (<0.001)
Skill ^‡^	13.98 ± 3.56	24.91 ± 2.46	−4.70 (<0.001)
Hand hygiene	2.69 ± 1.56	4.53 ± 0.61	−4.25 (<0.001)
Safe injection practice	1.04 ± 1.06	3.24 ± 1.59	−4.22 (<0.001)
Disinfection and environmental control	2.64 ± 1.17	4.46 ± 1.15	−4.42 (<0.001)
Risk assessment ang multidrug-resistant strains control	3.34 ± 1.11	4.86 ± 0.44	−4.45 (<0.001)
Urinary tract infection control	2.05 ± 1.49	4.02 ± 1.24	−4.38 (<0.001)
Respiratory infection and visitor control	2.21 ± 1.29	3.79 ± 1.21	−3.93 (<0.001)
Awareness of infection control ^§^	168.90 ± 8.74	171.10 ± 7.36	−1.77 (0.077)

* Wilcoxon signed-rank test. ^†^ Perfect Score: 100 points; ^‡^ Perfect Score: 30 points; ^§^ Perfect Score: 175 points. DACUM, Developing A Curriculum; M, mean; SD, standard deviation.

## Data Availability

Not applicable.

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
