# Peer review of "Development and Application of an Educational-Training Programme for Infection Control Practitioners in Long-Term Care Hospitals"

_healthcare, 2023, doi:10.3390/healthcare11040542_

Round 1
Reviewer 1 Report
Dear authors,
This is a really interesting and relevant study. I have a couple of minor typos/language items for you to check and a couple of suggestions/questions. These are as follows:
- In the materials and methods section, line 73: should this state 10-20 rather than 10~20?
- In the results section, line 130: should this state "...the scores indicating..." rather than "...he scores indicating..."?
- In table 2, under selecting disinfectants and identifying and advising how to disinfect medical equipment: there are a number of items that relate to risk e.g. classifying the risk of equipment and determining the level of disinfection according to the risks of the disinfection target. Do you mean risk here or should this be replaced with hazard?
- In 3.4 Educational-training program development: Unless used for an American audience or related to computing, program is usually spelt programme. So, please double check the spelling requirement for this journal. Also under this section, line 156, you sate "practice contents", if this is meant for rehearsal aspects of the course then practice should be spelt practise. This occurs again in the discussion at line 223. Please, note these are very minor points, just for checking.
- One final comment from me, although some aspects of infection prevention and control might have been deemed not difficult by the participants, this doesn't necessarily indicate competence or compliance. It would be nice for a future study to check/compare actual competence and compliance where possible with that which is perceived. I hope this makes sense.
I look forward to seeing your future research.
Author Response
Dear Reviewer:
We would like to thank you for taking the time to review our article. We have made corrections in the manuscript after going over the your comments. We revised the manuscript and all changes were marked in red text. We hope that the revised manuscript will better meet your comments and requirements.
Point 1: In the materials and methods section, line 73: should this state 10-20 rather than 10~20?
Response 1: “~” was changed to “-” (Line 106).
Point 2: In the results section, line 130: should this state "...the scores indicating..." rather than "...he scores indicating..."?
Response 2: We changed “he scores indicating…” to. “the scores indicating…” (line 205).
Point 3: In table 2, under selecting disinfectants and identifying and advising how to disinfect medical equipment: there are a number of items that relate to risk e.g. classifying the risk of equipment and determining the level of disinfection according to the risks of the disinfection target. Do you mean risk here or should this be replaced with hazard?
Response 3 In Table 2, we were contemplating between the term hazard and risk but subsequently determined that the term risk was more appropriate; hence, we changed risk to infection risk for better clarity (line 228).
Point 4: In 3.4 Educational-training program development: Unless used for an American audience or related to computing, program is usually spelt programme. So, please double check the spelling requirement for this journal. Also under this section, line 156, you sate "practice contents", if this is meant for rehearsal aspects of the course then practice should be spelt practise. This occurs again in the discussion at line 223. Please, note these are very minor points, just for checking.
Response 4 We changed “program” and “practice” to “programme” and “practise” (throughout the entire text, including lines 234 and 302).
Point 5: One final comment from me, although some aspects of infection prevention and control might have been deemed not difficult by the participants, this doesn't necessarily indicate competence or compliance. It would be nice for a future study to check/compare actual competence and compliance where possible with that which is perceived. I hope this makes sense.
Response 5 In Table 1, the difficulty level in 50 of the 51 tasks was 3 to 4 points out of 5 points, with higher scores indicating greater task difficulty. In this study, infection control practitionersfound most infection control tasks to be slightly difficult. To clarify the subjects' interpretation of difficulty according to task in Table 1, the following contents were included in the discussion (lines 329–335).
“of the 51 tasks, 50 had a difficulty level of between 3 and 4 points, indicating that most of the infection control tasks were recognized as slightly difficult by infection control practitioners. This is consistent with evidence showing that the knowledge evaluation score prior to the operation of the educational-training programme developed based on task analysis was slightly lower, between 2 and 3 points, in most items. Therefore, efforts to increase knowledge and competence through educational-99training programmes are necessary.”
Reviewer 2 Report
This study developed an infection control educational-training program for ICPs working in LTCHs based on a job analysis that utilizes the DACUM method, with the aim of improving work performance of ICPs in LTCHs. The subject of study of this paper is very timely given the aging of the population in today's societies. This article has also been well organized. Some commends would be provide for the authors as follows:
-
In the introduction section, the authors have not provided explanations of the reasons for choosing the DACUM method. Also, the authors need to provide more theories or literature to support the program they developed.
-
In the method section, it is too short and not clear enough. The authors should increase more details about the DACUM method and why to choose the research tools or variables (Job Analysis, Program Satisfaction, Knowledge and Skills, ...etc.).
-
In the result section, the authors tried their efforts to present the data organized tables. However, I would suggest to increase more texts to explain the tables. For example, I feel confused about how the authors concentrated on the nine key tasks in Table 2.
-
Conclusion section is a little bit too short and general. The research performed with a particular population and location. I would suggest highlighting the main research contributions of this study and also provide the suggestions for the future studies.
Author Response
Dear Reviewer:
We would like to thank you for taking the time to review our article. We have made corrections in the manuscript after going over the your comments. We revised the manuscript and all changes were marked in red text. We hope that the revised manuscript will better meet your comments and requirements.
Point 1: In the introduction section, the authors have not provided explanations of the reasons for choosing the DACUM method. Also, the authors need to provide more theories or literature to support the program they developed.
Response 1:
It was suggested in the previous manuscript that the DACUM method is effective in developing a training process for specific tasks because it is based on the job analysis of infection control practitioners in long-term care hospitals (lines 50-56). Reflecting your opinion, the need for this study was emphasized by adding the point that the infection control training programme developed previously was not developed based on job analysis (lines 57-74).
Point 2: In the method section, it is too short and not clear enough. The authors should increase more details about the DACUM method and why to choose the research tools or variables (Job Analysis, Program Satisfaction, Knowledge and Skills, ...etc.).
Response 2:
We Added DACUM workshop methods and procedures(lines 91-103).
Among the research variable names, 'Job Analysis' was judged to be inappropriate as a variable name because it is the process of deriving the duties and tasks of ICPs in LTCHs. Therefore, it was changed to ' The frequency, importance, and difficulty of the tasks of ICPs in LTCHs', and the reason why this variable was selected was added (lines 126-132). And the reason why 'ProgrammeSatisfaction' was selected as the research variable was to evaluate the process of the educational-training programme (lines 138-139). The reason why 'knowledge', 'skills' and 'awareness' were selected as study variables was to evaluate the effectiveness of the programme on learners (lines 145-146, 152-153). We have added the above contents to the manuscript.
Point 3: In the result section, the authors tried their efforts to present the data organized tables. However, I would suggest to increase more texts to explain the tables. For example, I feel confused about how the authors concentrated on the nine key tasks in Table 2.
Response 3
We have added text to explain Table 1 and Table 2 (lines 169-203). The process of deriving the nine key tasks was presented (lines 209-211, 215-218).
Point 4: Conclusion section is a little bit too short and general. The research performed with a particular population and location. I would suggest highlighting the main research contributions of this study and also provide the suggestions for the future studies.
Response 4
In the conclusion section, we emphasized the contributions of the study (lines 347-350) and added suggestions for future research (lines 355-358).
Reviewer 3 Report
Overview
Jeong and Kim present a study using the Developing A Curriculum (DACUM) method to design and pilot an educational training program for infection control practitioners in long-term care hospitals. It is wonderful to see how they carry out this data-driven program development, so this paper simultaneously presents their iterative process in building a program, while also providing the reader with details about what might be helpful and important for infection control practitioners in this setting to learn about. Please find specific comments below.
Abstract
· Line 10-11: I would recommend adding a qualifier to this as all elderly persons do not require LTCH stays, even if they have chronic illness or weaken immune systems.
Introduction
· Line 28-29: Would it be possible to clarify this a bit? It is unclear what the “rates of chronic illnesses increased to more than 90%” means. That 90% of the total population or total elderly population have a chronic illness? That 90% of the population will get a chronic illness at some point in their lives? That the rate of chronic illness increased by 90% compared to an earlier time period?
Methods
· Line 69-70: Was the one-day workshop with the 6-member DACUM committee or with another group?
· Can the authors add a date range to the methods section so it is clear when this work was conducted? That would be helpful for context, particularly given that the introduction explains there is increasingly more LTCHs across Korea.
· Line 67-69: Based on having the two citations after this sentence, I originally read this as the authors had conducted and published their literature review. However, the authors aren’t listed as authors for either of these citations, which makes me think that, instead, these were the only two papers they reviewed in their literature review. Given that, I would not consider use the language ‘literature review’ but would rather recommend saying that the authors based their duties examination on these two studies.
· Line 73: Would it be possible to clarify what the authors used as the region and hospital level designations? For instance, are these regions provinces or another designation?
· Line 82-83: How were participants selected for the study? Was this random or by convenience? Was it a subset of the ICPs who responded to the emailed survey or were participants selected from a different population?
· Line 83-84: The final participant number (29) should go in the results, not the methods section.
· Line 91-92: Please explain what the 5-point scale was for each of the frequency, importance, and difficulty measures. Relatedly, what did participants do in the survey if the duty was not part of their assigned duties (e.g. if someone else at the facility was in charge of that)?
Results
· Section 3.1: Based on this section, it sounds like the list itself is part of your study results. In that case, the author should move the first mention of the “51” tasks to here instead of line 90 in the methods.
· Section 3.1: Can any further details be provided in this section? It is very sparse. For instance, did the committee workshop reflect what was found in the literature or did the workshop bring to light anything that the literature missed?
· Line 129-130: Please review for spelling and verb vs. noun cohesiveness.
· Table 1: Can the authors add a footnote or other indication of what the 5-point scale represents? Currently, you have to read lines 129-132 and compare to the table or examine the logic of the ranking to understand if 5 is most important/difficult/frequent or is 1 is.
· Lines 127-128 and Table 1: It may be more helpful to readers to have the ‘totals’ by each of the 12 duties instead of across all tasks. I’m not sure how to interpret a total across all tasks collectively except as the latter key task cut offs.
· Line 134-135: The process of how key tasks for program development would be more appropriate in the methods section.
· Line 140-143: Check formatting and clarify who or what highlighted the importance of MDROs – the survey results or the workshop?
· Table 2: There’s some overlap between the elements of selecting disinfectants and identifying and advising how to disinfect medical equipment. Is the latter a subcategory of the former? Perhaps it would help to clarify in text or revise the elements to better distinguish.
· Table 2: Check influenza control capitalization
· Table 3: When did participants take the post-evaluation? Was that at the end of the 4th day?
· Section 3.6: How many points were the infection control knowledge and skills evaluations out of?
· The results do not outline participant characteristics at all even though section 2.4 says that the subjects’ general characteristics will be analyzed and that will help contextualize the results.
Discussion
· In table 1, it’s striking to see just how uniform the 51 tasks were in terms of difficulty (all but 1 is somewhere between 3-4). Why do the authors think that happened? Are all these truly about the same level of difficulty? Or was it a result of how the 5-levels were phrased? Or do people not want to seem either like their duties are too easy or that they are complaining? Some discussion of this could help readers better understand the results presented.
Author Response
Point 1: Overview
Jeong and Kim present a study using the Developing A Curriculum (DACUM) method to design and pilot an educational training program for infection control practitioners in long-term care hospitals. It is wonderful to see how they carry out this data-driven program development, so this paper simultaneously presents their iterative process in building a program, while also providing the reader with details about what might be helpful and important for infection control practitioners in this setting to learn about. Please find specific comments below.
Response 1: Thank you very much for the positive feedback on the development process of the educational-training program.
Point 2: Abstract
· Line 10-11: I would recommend adding a qualifier to this as all elderly persons do not require LTCH stays, even if they have chronic illness or weaken immune systems.
Response 2: In accordance with your comment, the following description was added: “All elderly persons do not require hospitalization in LTCHs, even if they have chronic illness or weaken immune systems, but ~” (lines 10–12).
Point 3: Introduction
· Line 28-29: Would it be possible to clarify this a bit? It is unclear what the “rates of chronic illnesses increased to more than 90%” means. That 90% of the total population or total elderly population have a chronic illness? That 90% of the population will get a chronic illness at some point in their lives? That the rate of chronic illness increased by 90% compared to an earlier time period?
Response 3
The sentence was modified as follows to clarify the contents: “Among the elderly aged 65 years or older, the prevalence of chronic illnesses diagnosed by a doctor was 89.2%” (lines 29–30).
Point 4: Methods
· Line 69-70: Was the one-day workshop with the 6-member DACUM committee or with another group?
Response 4
We added that the one-day workshop was held with the 6-member DACUM committee (lines 89-90).
Point 5: Can the authors add a date range to the methods section so it is clear when this work was conducted? That would be helpful for context, particularly given that the introduction explains there is increasingly more LTCHs across Korea.
Response 5
We added DACUM workshop (line 89), survey (line 119), and eucational-training program operation (line 120) date.
Point 6: Line 67-69: Based on having the two citations after this sentence, I originally read this as the authors had conducted and published their literature review. However, the authors aren’t listed as authors for either of these citations, which makes me think that, instead, these were the only two papers they reviewed in their literature review. Given that, I would not consider use the language ‘literature review’ but would rather recommend saying that the authors based their duties examination on these two studies
Response 6
We modified the wording according to your recommendation (lines 88-89).
Point 7: Line 73: Would it be possible to clarify what the authors used as the region and hospital level designations? For instance, are these regions provinces or another designation?
Response 7
The sentence was corrected (lines 105 to 108), saying that 10 to 20 LTCHs were selected for each of the 17 metropolitan local governments in Korea and evenly selected according to the size of beds.
Point 8: Line 82-83: How were participants selected for the study? Was this random or by convenience? Was it a subset of the ICPs who responded to the emailed survey or were participants selected from a different population?
Response 8
It was added that applicants for educational-training programme were conveniently recruited from ICPs who responded to the survey sent by e-mail (lines 118-119).
Point 9: Line 83-84: The final participant number (29) should go in the results, not the methods section.
Response 9
It was presented in the result part that 29 out of 30 people participated in the educational-training programme (line 252).
Point 10: Line 91-92: Please explain what the 5-point scale was for each of the frequency, importance, and difficulty measures. Relatedly, what did participants do in the survey if the duty was not part of their assigned duties (e.g. if someone else at the facility was in charge of that)?
Response 10 The 5-point scale provided scores ranging from 1 to 5 for frequency, importance, and difficulty (higher points indicated greater frequency, importance, and difficulty). Therefore, if the duty was not assigned among the surveyed subjects, we assigned a score of 1 point for frequency of performance. Thus, we added the following discussion according to the reviewer’s comments.
“The degree of frequency, importance, and difficulty was measured for each question on a 5-point scale ranging from 1 to 5 (with higher points indicating greater frequency, importance, and difficulty)” (Lines 133–135).
An explanation of the score was also added at the bottom of Table 1 (lines 223–224).
Point 11: Results
· Section 3.1: Based on this section, it sounds like the list itself is part of your study results. In that case, the author should move the first mention of the “51” tasks to here instead of line 90 in the methods.
Response 11
In the method section, ‘51’ was deleted (line 131).
Point 12: Section 3.1: Can any further details be provided in this section? It is very sparse. For instance, did the committee workshop reflect what was found in the literature or did the workshop bring to light anything that the literature missed?
Response 12
At the DACUM workshop, the job of ICPs in LTCHs in Korea was identified as 12 duties and 51 tasks. In general, these duties and tasks were similar to those of the ICP presented in references reviewed by the researcher prior to the workshop. This is described in the discussion section (lines 272-277). Corrected wording in Results section to reduce confusion (line 169).
Point 13: Line 129-130: Please review for spelling and verb vs. noun cohesiveness.
Response 13
We changed ‘he scores indicating~’ to ‘’the scores indicating~’(line 205).
Point 14: Table 1: Can the authors add a footnote or other indication of what the 5-point scale represents? Currently, you have to read lines 129-132 and compare to the table or examine the logic of the ranking to understand if 5 is most important/difficult/frequent or is 1 is.
Response 14
We added the following descriptions of the 5-point scale as a footnote to Table 1: a5-point scale, 1 = rarely, 2 = sometime, 3 = neutral, 4 = often, 5 = very often; b5-point scale, 1 = very low, 2 = low, 3 = neutral, 4 = high, 5 = very high; c5-point scale, 1 = very low, 2 = low, 3 = neutral, 4 = high, 5 = very high (lines 223–224).
Point 15: Lines 127-128 and Table 1: It may be more helpful to readers to have the ‘totals’ by each of the 12 duties instead of across all tasks. I’m not sure how to interpret a total across all tasks collectively except as the latter key task cut offs. Lines 127-128 and Table 1.
Response 15
Table 1 summarizes the results that form the basis for selecting the key tasks of the infection control practitioners for educational training programme development. We believe that the sum of tasks for each of the 12 duties in this table was not necessary for readers to interpret the contents of the table. Moreover, we determined that the readers may conflate duty with task; hence, we did not present the sum of tasks for each duty. Also, since the sum of all tasks is not needed, we deleted this from Table 1 (above line 223).
Point 16: Line 134-135: The process of how key tasks for program development would be more appropriate in the methods section.
Response 16
Added a procedure for deriving key tasks for program development in the Methods section (lines 111-113). However, the modifications as recommended by reviewer 2 are left in the results section (lines 209-211).
Point 17: Line 140-143: Check formatting and clarify who or what highlighted the importance of MDROs – the survey results or the workshop?
Response 17
Although the frequency of the methicillin-resistant Staphylococcus aureus (MRSA) infection control task was lower than the average, researchers selected it as a key task for the development of an educational-training program, considering the increased isolation of multidrug-resistant organisms (MDROs) in LTCHs in Korea. It is described in the text (lines 215-218).
Point 18: Table 2: There’s some overlap between the elements of selecting disinf. ectants and identifying and advising how to disinfect medical equipment. Is the latter a subcategory of the former? Perhaps it would help to clarify in text or revise the elements to better distinguish.
Response 18 The key task of “selecting disinfectants” is an element for selecting a disinfectant according to the risk of infection, whereas “identifying and advising how to disinfect medical equipment” is an element for identifying and advising on how to disinfect medical equipment. We deleted and modified redundant words and sentences to clearly distinguish the contents (Table 1).
Point 19: Table 2: Check influenza control capitalization
Response 19 We changed ‘Influenza’ to ‘influenza’ (Table 2).
Point 20: Table 3: When did participants take the post-evaluation? Was that at the end of the 4th day?
Response 20
Added to the Methods section that the post-evaluation time point was immediately after completion of the 4-day educational-training program (lines 121-122 and 123-124).
Point 21: Section 3.6: How many points were the infection control knowledge and skills evaluations out of?
Response 21
It was added that the perfect score for knowledge and skills is 30 points. (lines 257-258, lines 260)
Point 22: The results do not outline participant characteristics at all even though section 2.4 says that the subjects’ general characteristics will be analyzed and that will help contextualize the results.
Response 22
Given that the study results did not include the general characteristics of the subjects, Section 2.4 was modified to match the context with the study results (line 161).
Point 23: Discussion
In table 1, it’s striking to see just how uniform the 51 tasks were in terms of difficulty (all but 1 is somewhere between 3-4). Why do the authors think that happened? Are all these truly about the same level of difficulty? Or was it a result of how the 5-levels were phrased? Or do people not want to seem either like their duties are too easy or that they are complaining? Some discussion of this could help readers better understand the results presented.
Response 23 The researcher's interpretation and opinion on the difficulty level for each task presented in Table 1 were additionally presented in the discussion as follows(lines 330-336).
“of the 51 tasks, 50 had a difficulty level of between 3 and 4 points, indicating that most of the infection control tasks were recognized as slightly difficult by infection control practitioners. This is consistent with evidence showing that the knowledge evaluation score prior to the operation of the educational training programme developed based on task analysis was slightly lower, between 2 and 3 points, in most items. Therefore, efforts to increase knowledge and competence through educational training programmes are necessary.”
Round 2
Reviewer 2 Report
Dear author/authors,
Thank you for giving me the opportunity to review this valuable work. After my previous review and suggestions, I am glad to see that you have made positive changes without exception. I think positively.
With all respect.